# Prognostic Implication of EBV Infection in Gastric Carcinomas: A Systematic Review and Meta-Analysis

**DOI:** 10.3390/medicina59050834

**Published:** 2023-04-25

**Authors:** Jung-Soo Pyo, Nae-Yu Kim, Dong-Wook Kang

**Affiliations:** 1Department of Pathology, Uijeongbu Eulji Medical Center, Eulji University School of Medicine, Uijeongbu-si 11759, Republic of Korea; jspyo@eulji.ac.kr; 2Department of Internal Medicine, Uijeongbu Eulji Medical Center, Eulji University School of Medicine, Uijeongbu-si 11759, Republic of Korea; naeyu46@eulji.ac.kr; 3Department of Pathology, Chungnam National University Sejong Hospital, 20 Bodeum 7-ro, Sejong-si 30099, Republic of Korea; 4Department of Pathology, Chungnam National University School of Medicine, 266 Munhwa Street, Daejeon 35015, Republic of Korea

**Keywords:** gastric carcinoma, Epstein–Barr virus, overall survival, molecular classification, meta-analysis

## Abstract

*Background and objectives:* This study aims to elucidate the prognostic implications of Epstein–Barr virus (EBV) infection in gastric carcinomas (GCs) through a systematic review and meta-analysis. *Materials and Methods:* In total, 57 eligible studies and 22,943 patients were included in this meta-analysis. We compared the prognoses of EBV-infected and non-infected GC patients. The subgroup analysis was performed based on the study location, molecular classification, and Lauren’s classification. This study was checked according to the PRISMA 2020. The meta-analysis was performed using the Comprehensive Meta-Analysis software package. *Results:* EBV infection was found in 10.4% (95% confidence interval (CI) 0.082–0.131) of GC patients. The EBV-infected GC patients had a better overall survival compared with the EBV-non-infected GC patients (hazard ratio (HR) 0.890, 95% CI 0.816–0.970). In the subgroup analysis based on molecular classification, no significant differences were found between EBV+ and microsatellite instability and microsatellite stable (MSS)/EBV− subgroups (HR 1.099, 95% CI 0.885–1.364 and HR 0.954, 95% CI 0.872–1.044, respectively). In the diffuse type of Lauren’s classification, EBV-infected GCs have a better prognosis compared with the EBV-non-infected GCs (HR 0.400, 95% CI 0.300–0.534). The prognostic impact of EBV infection was found in the Asian and American subgroups but not in the European subgroup (HR 0.880, 95% CI 0.782–0.991, HR 0.840, 95% CI 0.750–0.941, and HR 0.915, 95% CI 0.814–1.028). *Conclusions:* EBV infection is a favorable survival factor for GCs. However, the prognostic implications of EBV infection in the new molecular classification are not clear.

## 1. Introduction

Gastric carcinoma (GC) is a common malignancy of the gastrointestinal tract and is a heterogeneous disease with different phenotypes, genotypes, and tumor behaviors [1,2]. GC carcinogenesis is associated with infectious agents, including *Helicobacter pylori* and Epstein–Barr virus (EBV). Burke et al. first reported a correlation between EBV infection and GC in 1990 [3], but the molecular classification of GC has only recently been introduced, with follow-up studies also being only recently published. Based on this molecular classification, GCs can be categorized as EBV-positive, microsatellite instability-high, genomically stable, and chromosomally instable. In the study by Burke et al., EBV-positive GC comprised 9% of overall GC cases. All EBV-positive tumors clustered together and exhibited an extreme CpG island methylator phenotype distinct from that in the microsatellite instability (MSI) subtype, which was consistent with the reports of previous studies. EBV-associated GC (EBVaGC) has since been associated with DNA hypermethylation, PIK3CA gene mutations, and AT-rich interactive domain-containing protein 1A (ARID1A) gene expression. The differences in tumor subtype and molecular composition can affect tumor behaviors [4,5,6,7,8]. The EBV positivity rate in GCs can vary depending on various factors, such as region [2,4,5,6,7,8,9,10,11,12,13,14,15,16,17,18,19,20,21,22,23,24,25,26,27,28,29,30,31,32,33,34,35,36,37,38,39,40,41,42,43,44,45,46,47,48,49,50,51,52,53,54,55,56,57,58,59]. The prognostic implications of EBV positivity may differ depending on various factors, such as study location, Lauren’s classification, and molecular classification. Recently, EBV-positive GCs had better response for perioperative chemoradiotherapy [4]. In addition, EBV was correlated with tumor-infiltrating lymphocytes and tertiary lymphoid structures [5]. GC with lymphoid stroma (GCLS) is defined as undifferentiated gastric carcinoma with prominent lymphoid infiltration [43]. The EBV infection can be useful for predicting the prognosis of GCLS [43]. Thus, the prognostic role of EBV will be evaluated based on various factors, including tumor subtype and molecular composition. A meta-analysis can be useful in providing a detailed evaluation of the prognostic impact of EBV positivity. This study aims to identify the prognostic role of EBV infection according to the clinicopathological characteristics of GC patients. We evaluated the prognostic impact of EBV positivity in GCs using a meta-analysis. A series of subgroup analyses based on the study location, Lauren’s classification, and molecular classification were also performed. In addition, the prognostic implications of EBV were evaluated in GCLS and non-GCLS.

## 2. Materials and Methods

### 2.1. Published Study Search and Selection Criteria

Published articles were searched in the PubMed database on 15 August 2022. The keywords used in the search were “gastric carcinoma or gastric cancer or stomach cancer” and “Epstein–Barr virus or EBV”. The titles and abstracts of all searched articles were screened for the inclusion and exclusion of each article. Articles that studied the correlation between EBV infection and the overall survival of GCs were included. Case reports, non-original articles, or articles not written in English were excluded. This study was performed by the Preferred Reporting Items for Systematic Reviews and Meta-Analyses (PRISMA). The PRISMA checklist is shown in the Appendix A. In addition, this review was not registered.

### 2.2. Data Extraction

All data were obtained from eligible studies [2,4,5,6,7,8,9,10,11,12,13,14,15,16,17,18,19,20,21,22,23,24,25,26,27,28,29,30,31,32,33,34,35,36,37,38,39,40,41,42,43,44,45,46,47,48,49,50,51,52,53,54,55,56,57,58,59] by two independent authors (J.S.P. and N.Y.K.). The extracted data included the authors’ information, study location, number of analyzed patients, EBV positivity rates, and information about survival. EBV positivity rates were investigated according to study location and tumor location. For the subgroup analysis, information about survival was additionally gathered based on tumor subtypes, such as the presence of lymphoid stroma, Lauren’s classification, and molecular classification. For a quantitative aggregation of the survival results, we obtained the hazard ratio (HR) using one of the following methods. In studies not quoting the HR or its confidence interval (CI), these variables were calculated from the presented data using the HR point estimate, log-rank statistic or its *p*-value, and the O–E statistic (difference between the number of observed and expected events) or its variance. If those data were unavailable, HR was estimated using the total number of events, the number of patients at risk in each group, and the log-rank statistic or its *p*-value. Finally, if the only useful data were in the form of graphical representations of survival distributions, the survival rates were extracted at specified times to reconstruct the HR estimate and its variance under the assumption that patients were followed up at constant intervals [60]. The published survival curves were read independently by two researchers in order to reduce variability. The HRs were then combined into an overall HR using Peto’s method [61].

### 2.3. Statistical Analyses

The meta-analysis was performed using the Comprehensive Meta-Analysis software package (Biostat, Englewood, NJ, USA). The rates of EBV-infected GCs were evaluated in eligible studies, and the differences in overall survival between EBV-infected and non-infected GCs were investigated. Subgroup analyses for EBV positivity rates were performed based on study and tumor locations. In survival analysis, subgroup analyses were performed based on Lauren’s classification and molecular classification. In addition, the overall survival between patients with EBV-infected GCLS and those with non-GCLS were compared. We checked the heterogeneity between the studies based on the Q and I^2^ statistics, expressed as *p*-values. Additionally, we conducted a sensitivity analysis to assess the heterogeneity of the eligible studies and the impact of each study on the combined effects. In the meta-analysis, as the eligible studies used a variety of different populations, a random effects model (rather than a fixed effects model) was determined to be more suitable. In the case of 1 included literature, the *p*-value of heterogeneity was 1. In addition, the effect size of the fixed effect and the random effect model is the same. The statistical difference between subgroups was evaluated with a meta-regression test. We used Begg’s funnel plot and Egger’s test to assess the publication bias; if significant publication bias was found, the fail-safe N and trim-fill tests were additionally used to confirm the degree of publication bias. The results were considered statistically significant at *p* < 0.05.

## 3. Results

### 3.1. Selection and Characteristics of the Studies

Through a database search, 1500 articles were obtained. During their review, 1443 articles were excluded. Five hundred fifty-nine articles lacked sufficient information. Of the remaining articles, 385 articles were excluded as they were non-original articles. Another 499 articles were excluded; the reasons for this include non-human studies (*n* = 256), articles on other diseases (*n* = 195), written in a language other than English (*n* = 46), and duplication (*n* = 2). Finally, 57 eligible articles were selected and included in the meta-analysis (Figure 1 and Table 1). These studies included 22,943 patients with GC.

### 3.2. Meta-Analysis

First, the EBV positive rates of GCs were investigated and analyzed through a meta-analysis. The estimated EBV positivity rate of GCs was 0.104 (95% CI 0.082–0.131; Table 2). The *p*-value of the heterogeneity test was less than 0.001. In addition, there was no significant publication bias in an Egger’s test (*p* = 0.860). In the subgroup analysis based on study location, the EBV positivity rates in the Asian, American, and European subgroups were 0.114 (95% CI 0.080–0.158), 0.107 (95% CI 0.079–0.142), and 0.084 (95% CI 0.070–0.100), respectively. In the subgroup analysis, there was no significant publication bias. The EBV positive rates of antrum, body, and cardia/fundus were 0.083 (95% CI 0.055–0.123), 0.130 (95% CI 0.095–0.176), and 0.182 (95% CI 0.126–0.257), respectively.

Comparisons of the overall survival between the patients with EBVaGC and those with non-EBVaGC were performed. Those with EBVaGC showed better overall survival than those with non-EBVaGC (HR 0.882, 95% CI 0.813–0.957; Table 3). The *p*-value of the heterogeneity test was less than 0.001. In addition, there was no significant publication bias in an Egger’s test (*p* = 0.297). Next, a subgroup analysis was performed based on the histologic subtype, Lauren’s classification, and molecular classification. There was a significant difference in the overall survival between the patients with EBVaGC and those with non-EBVaGC in GCLS, but not in non-GCLS (HR 0.178, 95% CI 0.059–0.537 and HR 0.870, 95% CI 0.620–1.222, respectively). The Egger’s test was not performed due to the number of included articles in GCLS and non-GCLS subgroups. In the diffuse type of Lauren’s classification, the patients with EBVaGC had better prognosis than those with non-EBVaGC (HR 0.400, 95% CO 0.300–0.534). However, there was no significant difference in the overall survival between the patients with EBVaGC and those with non-EBVaGC in the intestinal type of Lauren’s classification (HR 1.364, 95% CI 0.425–0.4379). In addition, Egger’s test was not performed due to the number of included articles in intestinal and diffuse type subgroups. In the molecular classification, there were no significant differences in the overall survival between the patients with GCs who were EBV positive and MSI-high (MSI-H) and those with GCs who were microsatellite stable (MSS) and EBV negative (HR 1.099, 95% CI 0.885–1.364 and 0.954, 95% CI 0.872–1.044, respectively). There was no significant publication bias in this subgroup analysis. In the Asian and American subgroups, the patients with EBVaGC showed better prognosis than those with non-EBVaGC (HR 0.880, 95% CI 0.782–0.991 and HR 0.840, 95% CI 0.750–0.941, respectively; Figure 2A,B). However, there were no significant correlations between EBV infection and overall survival in the European subgroup (0.915, 95% CI 0.814–1.028; Figure 2C).

## 4. Discussion

The estimated EBV positivity rate in the present study was 10.4%. In the eligible studies, the EBV positivity rates ranged from 2.0% to 40.1% [2,4,5,6,7,8,9,10,11,12,13,14,15,16,17,18,19,20,21,22,23,24,25,26,27,28,29,30,31,32,33,34,35,36,37,38,39,40,41,42,43,44,45,46,47,48,49,50,51,52,53,54,55,56,57,58,59]. A meta-analysis is useful for elucidating the prognostic role of EBV in GCs as it has a diverse distribution of EBV positivity. To the best of our knowledge, this meta-analysis was the first to evaluate the prognostic role of EBV positivity based on the molecular classification of GCs. Our results show that the prognostic impact of EBV differed between the overall cases and the molecular classification.

Although the estimated EBV positivity rate was 10.4%, the EBV positivity rate in the eligible studies reached 40.1%. In addition, the EBV positivity rates differed among the various study locations. Therefore, it is important to evaluate the prognostic impact of EBV positivity in various subgroups. In our previous study, EBV infection significantly correlated with sex, GCLS, programmed death ligand-1 expression, and tumor-infiltrating lymphocytes [62]. In another meta-analysis, a correlation between EBV infection and GC prognosis was reported [63]. Liu et al. reported that EBV was significantly correlated with a favorable prognosis of GCs, which was similar to our results [63]. However, this meta-analysis was conducted in 2015 and included eligible studies published from 1996 to 2014 [63]. Another previous meta-analysis included 24 eligible studies. Furthermore, the present meta-analysis included 57 eligible studies. In addition, a subgroup analysis based on molecular classification was not performed [63]. Recently, the molecular classification of GCs has been introduced and investigated [1]. As EBV infection is important in this molecular classification, a detailed evaluation based on molecular classification is needed.

Previous studies have reported that the patients with EBVaGC had a better prognosis compared with those with non-EBVaGC. In the present meta-analysis, EBVaGC was significantly correlated with better prognosis compared with non-EBVaGC (HR 0.890, 95% CI 0.816–0.970). We additionally performed a subgroup analysis based on various factors such as study location, Lauren’s classification, and molecular classification. In the subgroup analysis based on the study location, statistical significance was found in the Asian and American subgroups, but not in the European subgroup. This result was similar to that of a previous meta-analysis [63]. According to Lauren’s classification, intestinal-type GCs had more favorable survival than diffuse-type GCs [64,65]. The prognostic impacts were also identified in the diffuse type of Lauren’s classification and GCLS (HR 0.400, 95% CI 0.300–0.534 and HR 0.178, 95% CI 0.059–0.537, respectively). Koriyama et al. reported the prognostic significance of EBV involvement according to Lauren’s classification [33]. They suggested that the opposite prognostic impact of EBV was identified in the diffuse and intestinal types of Lauren’s classification. Interestingly, a favorable prognosis was identified in the diffuse type, but not in the intestinal type, according to Lauren’s classification. However, our results differ from those of a previous study.

The different tumor behaviors of GCs can be affected by their histologic and molecular characteristics [1,66], and recently, a molecular classification of GCs has been introduced [1]. The patients with EBV-positive or MSI-high GC had better prognosis compared with those with EBV-negative or MSS GC [1]. Kohlruss et al. analyzed prognosis according to the new molecular classification, but no statistical differences were found between the molecular subgroups [67]. The prognostic impact of EBV according to this molecular classification is unclear in individual studies. The present study also performed a subgroup analysis based on this molecular classification. No significant difference was observed in the prognosis between the EBV positive and MSI-H subgroup and the MSS and EBV negative subgroup (HR 1.099, 95% CI 0.885–1.364 and HR 0.954, 95% CI 0.872–1.044, respectively). Since the nature of the tumor may differ depending on the molecular classification, the therapeutic effect of chemotherapy may also vary. Hence, further studies are needed to determine the prognostic role of EBV according to the molecular subtypes or GC therapy.

The patients with EBVaGC have better prognosis compared with those with non-EBVaGC. Similar results were obtained in our meta-analysis. However, individual studies reported different results, while some showed contrasting results. The intestinal type of EBVaGC had a worse prognosis than the intestinal type of non-EBVaGC [33]. In addition, Lin et al. reported a correlation between EBV and worse prognosis [38]. Interestingly, several studies have shown no significant difference in the survival rates between the patients with EBVaGC and those with non-EBVaGC. A correlation between EBV positivity and better prognosis was reported in a few eligible studies [17,18,33,40,42,45,46,51,58]. Furthermore, other studies have shown different results regarding the prognostic significance of EBV positivity. Therefore, this meta-analysis may be useful for interpreting the prognosis of EBVaGC. In a previous meta-analysis, EBV positivity was significantly correlated with sex [62]. The factors associated with tumor progression and behavior were not correlated with EBV positivity [62]. However, in the present study, EBVaGC was significantly correlated with favorable prognosis. A significant correlation was also found between EBV and GCLS levels. We also compared the prognostic impact of EBV infection between the patients with GCLS and those without GCLS. EBV was associated with a favorable prognosis in the patients with GCLS but not in those with non-GCLS (HR 0.178, 95% CI 0.059–0.537 and HR 0.870, 95% CI 0.620–1.222, respectively). To understand this discrepancy, an analysis of the results of the subgroup analysis can be helpful.

This study has some limitations. First, some studies show that EBVaGC is associated with worse prognosis. This discrepancy might be due to the different populations evaluated and methods used for detecting EBV. The significance of the prognostic implication of EBV was reported in 7 out of 32 Asian studies, 1 out of 11 American studies, and 1 out of 12 European studies. The individual studies had sample sizes ranging from 43 to 1328 participants. The EBV positivity ranged from 2.0% to 40.1%. As the individual studies included a variety of different populations, the results may vary between eligible studies. However, our meta-analysis did not confirm this cause. Second, differences were found in the number of studies conducted before and after the introduction of the molecular classification. The amount of analyzed results according to the molecular classification were too few compared with the total number of analyzed results. Third, the patients with GCLS and those with diffuse-type GC and EBV infection had better prognosis. However, since the number of studies presenting this information is too small, further studies are warranted. Fourth, the EBV positive rates were differed by the tumor location. However, we could not perform the detailed analysis for the prognostic impacts of EBV based on the tumor location due to insufficient information. Further study on the impact of the tumor location will be useful for predicting the prognostic role of EBV.

## 5. Conclusions

Although EBV infection was significantly correlated with favorable survival in the patients with GCs, the details of this patient group differed according to the new molecular classification and by subtype. However, this meta-analysis may still be considered significant as the prognostic role of EBV may vary depending on the GC subtype. Hence, further detailed studies on the subtypes with fewer cases are needed.

## Figures and Tables

**Figure 1 medicina-59-00834-f001:**
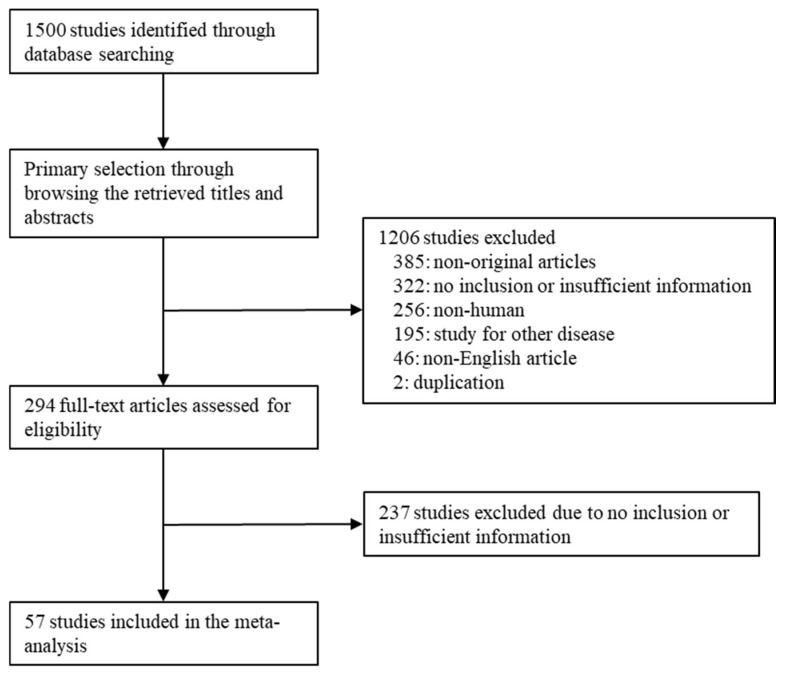
Flow chart of the searching strategy.

**Figure 2 medicina-59-00834-f002:**
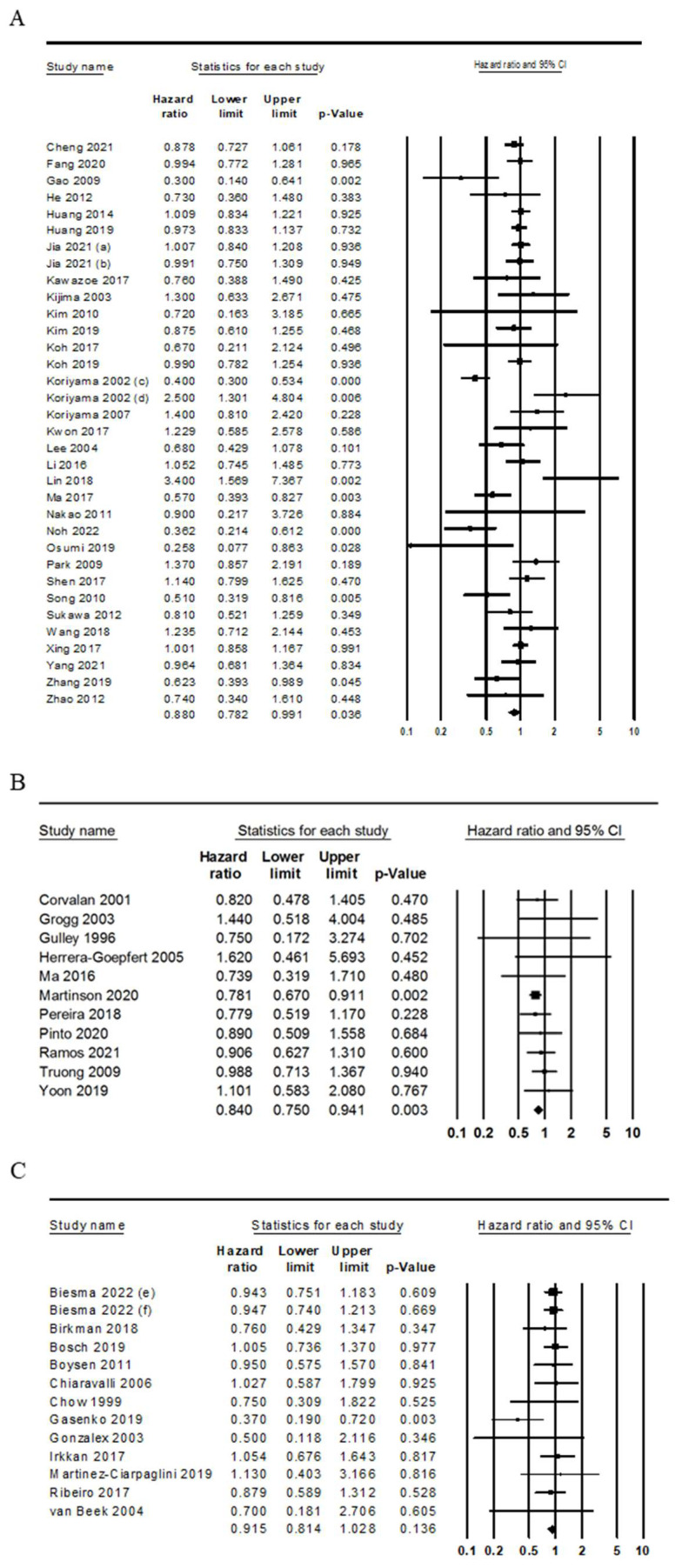
Subgroup analysis based on the study location. (**A**) Asia. (**B**) America. (**C**) Europe. (a, No chemotherapy; b, chemotherapy; c, diffuse type; d, intestinal type; e, CRITICS trial; f, D1/D2 trial).

**Table 1 medicina-59-00834-t001:** Main characteristics of the eligible studies.

First Author, Year	Location	No. of Patients	No. of EBV+	First Author, Year	Location	No. of Patients	No. of EBV+
Biesma 2022 [4]	Netherlands			Koriyama 2007 [34]	Japan	149	49
(CRITICS Trial) (D1/D2 Trial)		454 447	25 47	Kwon 2017 [35]	Korea	394	26
Birkman 2018 [9]	Finland	186	17	Lee 2004 [36]	Korea	1114	63
Bösch 2019 [10]	Germany	189	11	Li 2016 [37]	China	137	30
Boysen 2011 [11]	Denmark	186	18	Lin 2018 [38]	China	217	87
Cheng 2021 [5]	China	846	42	Ma 2016 [39]	USA	44	7
Chiaravalli 2006 [12]	Italy	113	17	Ma 2017 [40]	China	571	31
Cho 2018 [13]	Korea	58	29	Martinez-Ciarpaglini 2019 [41]	Spain	209	13
Chow 1999 [14]	Poland	87	11	Martinson 2020 [42]	USA	82	19
Corvalan 2001 [15]	Chile	145	27	Min 2016 [43]	Korea	145	124
Fang 2020 [16]	Taiwan	460	43	Nakao 2011 [44]	Korea	371	20
Gao 2009 [17]	China	1039	21	Noh 2022 [45]	Korea	956	65
Gasenko 2019 [18]	Latvia	302	26	Osumi 2019 [46]	Japan	898	71
Gonzalex 2003 [19]	Various, Europe	87	4	Park 2009 [47]	Korea	457	50
Grogg 2003 [20]	USA	110	7	Pereira 2018 [48]	Brazil	286	30
Gulley 1996 [21]	USA	95	11	Pinto 2020 [6]	Chile	91	12
He 2012 [22]	China	118	21	Ramos 2021 [7]	Brazil	287	30
Herrera-Goepfert 2005 [23]	Mexico	135	8	Ribeiro 2017 [49]	Turkey	179	15
Huang 2014 [24]	Taiwan	1020	52	Shen 2017 [50]	China	202	42
Huang 2019 [2]	Taiwan	1248	65	Song 2010 [51]	Korea	528	123
Irkkan 2017 [25]	Turkey	105	8	Sukawa 2012 [52]	Japan	222	18
Jia 2021 [26]	China	1328	55	Truong 2009 [53]	USA	235	12
Kawazoe 2017 [27]	Japan	487	25	van Beek 2004 [54]	Netherlands	566	41
Kijima 2003 [28]	Korea	420	28	Wang 2018 [55]	China	147	35
Kim 2010 [29]	Korea	247	18	Xing 2017 [56]	Portugal	966	33
Kim 2019 [30]	USA	43	6	Yang 2021 [8]	China	226	13
Koh 2017 [31]	Korea	392	25	Yoon 2019 [57]	USA	107	3
Koh 2019 [32]	Korea	894	79	Zhang 2019 [58]	China	1013	58
Koriyama 2002 [33]	Japan	192	64	Zhao 2012 [59]	China	711	80

**Table 2 medicina-59-00834-t002:** Estimated Epstein–Barr virus-infected rates in gastric carcinomas.

	Number of Subsets	Fixed Effect (95% CI)	Heterogeneity Test (*p*-Value)	Random Effect (95% CI)	Egger’s Test (*p*-Value)
Overall	55	0.103 (0.099, 0.108)	<0.001	0.104 (0.082, 0.131)	0.860
Location					
Asia	31	0.106 (0.101, 0.111)	<0.001	0.114 (0.080, 0.158)	0.484
America	12	0.115 (0.100, 0.132)	<0.001	0.107 (0.079, 0.142)	0.256
Europe	12	0.084 (0.075, 0.095)	0.026	0.084 (0.070, 0.100)	0.667

CI, confidence interval.

**Table 3 medicina-59-00834-t003:** Comparison of overall survival between Epstein–Barr virus-infected and non-infected gastric carcinoma.

	Number of Subsets	Fixed Effect (95% CI)	Heterogeneity Test (*p*-Value)	Random Effect (95% CI)	Egger’s Test (*p*-Value)
Overall	59	0.901 (0.861, 0.944)	<0.001	0.882 (0.813, 0.957)	0.297
GCLS	2	0.178 (0.059, 0.537)	0.799	0.178 (0.059, 0.537)	NA
Non-GCLS	1	0.870 (0.620, 1.222)	1.000	0.870 (0.620, 1.222)	NA
Lauren’s classification					
Intestinal type	2	1.274 (0.829, 1.960)	0.007	1.364 (0.425, 4.379)	NA
Diffuse type	1	0.400 (0.300, 0.534)	1.000	0.400 (0.300, 0.534)	NA
Molecular classification					
EBV+ vs. MSI high	12	1.099 (0.885, 1.364)	0.932	1.099 (0.885, 1.364)	0.426
EBV+ vs. MSS/EBV−	12	0.954 (0.872, 1.044)	0.967	0.954 (0.872, 1.044)	0.107

CI, confidence interval; EBV, Epstein–Barr virus; MSI, microsatellite instability; MSS, microsatellite stable; GCLS, gastric carcinoma with lymphoid stroma.

## Data Availability

Not applicable.

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
