# Peer review of "Prognostic Implication of EBV Infection in Gastric Carcinomas: A Systematic Review and Meta-Analysis"

_medicina, 2023, doi:10.3390/medicina59050834_

Round 1
Reviewer 1 Report
This is a large meta-analysis covering an important, but insidious topic. The statistics is appropriate. The main defect is in English language and in frequent redundance that make it hard to comprehend and to follow the development of reasoning. This is true in particular for the discussion, in which, for example, is frequently repeated the need of a meta-analysis.
Specific remarks:
- Line 20: The term "overall" is repeated three times in only one line
- Line 48-49: Substitute the preposition "Therefore"
- Figure 1: The sum of studies excluded in the primary selection reaches 1206, not 1271
- Table 1: Check carefully the total number of patients
- Results: It might be interesting to evalute the frequence of HBV+ in relation to tumor site (body, fundus, antrum, etc)
- Table 3: There is a reason for the fact that only in the sub-group "intestinal type" the random effect is different from the "fixed effect"? Please, explain.
- Fig. 2: Some confusion arises because the notation indicating specific characteristic of studies (chemotherapy, no chemotherapy, diffuse type, intestinal type, etc) uses the same alphabetical letters that indicate the three part of the figure (A, B, C). Please, use different symbols or, at least, use lowercase characters.
Author Response
This is a large meta-analysis covering an important, but insidious topic. The statistics is appropriate. The main defect is in English language and in frequent redundance that make it hard to comprehend and to follow the development of reasoning. This is true in particular for the discussion, in which, for example, is frequently repeated the need of a meta-analysis.
Specific remarks:
- Line 20: The term "overall" is repeated three times in only one line
Response:
As recommendation, we corrected the sentences.
- Line 48-49: Substitute the preposition "Therefore"
Response:
As recommendation, we corrected the sentence.
- Figure 1: The sum of studies excluded in the primary selection reaches 1206, not 1271
Response:
As recommendation, we corrected the number of articles and replaced figure 1. In addition, the description of the number was corrected in the result part.
- Table 1: Check carefully the total number of patients
Response:
As a recommendation, we corrected the total number of patients from 22,668 to 22,943.
- Results: It might be interesting to evalute the frequence of EBV+ in relation to tumor site (body, fundus, antrum, etc)
Response:
As a recommendation, we analyzed the EBV-positive rate based on tumor location. The EBV-positive rates of antrum, body, and cardia/fundus were respectively 0.083 (95% CI
0.055-0.123), 0.130 (95% CI 0.095-0.176), and 0.182 (95% CI 0.126-0.257). We added this result in the revised manuscript.
- Table 3: There is a reason for the fact that only in the sub-group "intestinal type" the random effect is different from the "fixed effect"? Please, explain.
Response:
In the case of one included literature, the effect size of the Fixed effect and the random effect model is the same.
- Fig. 2: Some confusion arises because the notation indicating specific characteristic of studies (chemotherapy, no chemotherapy, diffuse type, intestinal type, etc) uses the same alphabetical letters that indicate the three part of the figure (A, B, C). Please, use different symbols or, at least, use lowercase characters.
Response:
As a recommendation, we changed the symbols.
* We performed extensive editing of English language and style by MDPI.

Reviewer 2 Report
The authors aimed to assess the prognostic implications of EBV infection in patients with gastric cancer. I have several comments:
TITLE
· Please change the title to: "Prognostic implication of Epstein-Barr virus infection in patients with Gastric Carcinomas: A systematic review meta-analysis"
ABSTRACT
· You should write each section of Background, Methods, Results, and Conclusions separately.
· Please write the type of studies included.
· Please state the statistical tests used in the Methods.
· What do you mean by survival? In other words, did you estimate the survival by hazard ratio?
INTRODUCTION
· You should add more studies relevant to your topic recently published.
· Please state the clinical impressions of your study in the last paragraph of the Introduction. What problems remain unanswered? What are questions responding to?
METHODS
· Please report your study according to the PRISMA guideline.
RESULTS
· How many articles were finally included? 55 or 57?
DISCUSSION
· How can the high heterogeneity be explained?
· Please compare your results with this meta-analysis: https://www.ncbi.nlm.nih.gov/pmc/articles/PMC4619309/
Author Response
The authors aimed to assess the prognostic implications of EBV infection in patients with gastric cancer. I have several comments:
TITLE
- Please change the title to: "Prognostic implication of Epstein-Barr virus infection in patients with Gastric Carcinomas: A systematic review meta-analysis"
Response:
As a recommendation, we changed the Title.
ABSTRACT
- You should write each section of Background, Methods, Results, and Conclusions separately.
Response:
As a recommendation, we added the section.
- Please write the type of studies included.
Response:
As a recommendation, we added the type of study.
- Please state the statistical tests used in the Methods.
Response:
As a recommendation, we added the statement.
- What do you mean by survival? In other words, did you estimate the survival by hazard ratio?
Response:
Included studies evaluated the prognostic role of EBV infection by a hazard ratio of EBV-infected and non-infected groups.
INTRODUCTION
- You should add more studies relevant to your topic recently published.
Response:
As a recommendation, we revised the references.
- Please state the clinical impressions of your study in the last paragraph of the Introduction. What problems remain unanswered? What are questions responding to?
As a recommendation, we added the explanation of the questions of this study in the revised manuscript.
METHODS
- Please report your study according to the PRISMA guideline.
Response:
As a recommendation, we described PRISMA checklist in the revised manuscript and added the supplementary data (S1).
RESULTS
- How many articles were finally included? 55 or 57?
Response:
The number of included studies in the present meta-analysis was 57. We corrected the typo-error.
DISCUSSION
- How can the high heterogeneity be explained?
Response:
We thought that heterogeneity was high due to the diversity of the population group and laboratory differences. Thus, various subgroup analyses were performed to identify the cause of high heterogeneity.
- Please compare your results with this meta-analysis: https://www.ncbi.nlm.nih.gov/pmc/articles/PMC4619309/
Response:
As a recommendation, we described in the original manuscript as below:
In another meta-analysis, a correlation between EBV infection and GC prognosis was reported [63]. Liu et al. reported that EBV was significantly correlated with a favorable prognosis of GCs, which was similar to our results [63]. However, this meta-analysis was conducted in 2015 and included eligible studies published from 1996 to 2014 [63]. Another previous meta-analysis included 24 eligible studies. Meanwhile, the present meta-analysis included 57 eligible studies. In addition, a subgroup analysis based on molecular classification was not performed [63].
Previous studies have reported that EBVaGC patients had a better prognosis compared with non-EBVaGC patients. In the present meta-analysis, EBVaGC was significantly correlated with a better prognosis compared with non-EBVaGC (HR 0.890, 95% CI 0.816-0.970). We additionally performed a subgroup analysis based on various factors such as study location, Lauren’s classification, and molecular classification. In the subgroup analysis based on the study location, statistical significance was found in the Asian and American subgroups, but not in the Europe subgroup. This result was similar to that of a previous meta-analysis [63].
Reference
- Liu, X.; Liu, J.; Qiu, H.; Kong, P.; Chen, S.; Li, W.; Zhan, Y.; Li, Y.; Chen, Y.; Zhou, Z.; Xu, D.; Sunx X. Prognostic significance of Epstein-Barr virus infection in gastric cancer: a meta-analysis. BMC Cancer. 2015, 15, 782.
* We performed extensive editing of English language and style by MDPI.

Round 2
Reviewer 2 Report
Thanks for revisions.